# The Effect of Input Digitalization on Carbon Emission Intensity: An Empirical Analysis Based on China’s Manufacturing

**DOI:** 10.3390/ijerph20043174

**Published:** 2023-02-11

**Authors:** Luyang Tang, Bangke Lu, Tianhai Tian

**Affiliations:** 1School of Business Administration, Zhongnan University of Economics and Law, Wuhan 430073, China; 2School of Statistics and Mathematics, Zhongnan University of Economics and Law, Wuhan 430073, China; 3School of Mathematics, Monash University, Melbourne, VIC 3800, Australia

**Keywords:** input digitalization, carbon emission intensity, environmental benefit, manufacturing industry

## Abstract

Digitalization is an excellent opportunity for the manufacturing industry all over the world to improve the core competitiveness and break through the “low-end locking” dilemma. However, it is not clear whether the digitalization of the manufacturing industry has positive ecological and environmental benefits under the resource and environmental constraints. To answer this question, we use the data from the world input–output database (WIOD) to investigate the impact of manufacturing input digitalization on carbon emission intensity by an extended analysis. The results show that the input digitalization of the manufacturing industry has mixed effects on reducing carbon emission intensity. The productive input digitalization can reduce carbon emission intensity, but the distributional input digitalization may increase carbon emission intensity. Non-pollution-intensive manufacturing and high-input digital manufacturing have stronger carbon emission reduction effects than the other industry sectors. From the perspective of input sources, input digitalization from domestic sources has a significant inhibitory effect on the carbon emission intensity. In contrast, input digitalization from foreign sources may increase carbon emission intensity.

## 1. Introduction

Since the 1990s, the digital economy has gradually become an important driving force for economic transformation and upgrading in developed countries, such as the United States and countries in Europe. The business, transportation, trade, and other economic modes have undergone revolutionary changes [1,2,3]. Digital production is the process by which creative ideas and assets (e.g., images, as well as text and interactive apps) are translated into an array of digital media. Therefore, data, as a key factor of production, participate in the processes of international labor division and value creation. In recent years, the digital transformation of manufacturing has been the key for countries to compete for the new global economic and strategic heights. Compared with the international advanced level, China’s manufacturing industry is significant, but not robust. Thus, it is imperative for China to accelerate the integration of the advanced information technologies and manufacturing industries.

In the face of increasingly severe resource and environmental constraints, major economies have recognized the importance of green development. The United States proposes to achieve ‘net-zero emissions’ by 2050. The European Union has submitted the European Climate Law, which aims to legally ensure that Europe becomes the first ‘climate-neutral’ continent by 2050. China plans to achieve carbon peaking in 2030 and carbon neutrality in 2060. China’s extensive economic growth model comes at the expense of an environmental overdraft and ecological damage. At present, China is one of the countries with the highest carbon emissions in the world, of which industrial emissions account for about 68% of the total emission. However, energy requirement is still increasing and carbon emissions have not yet peaked. The realization of the structural transformation of the manufacturing industry and the transformation of low-carbon energy are urgent problems that China needs to solve currently [4].

Some scholars have explored the relationship between the digital economy and environmental problems [5,6]. The results show that the digital economy has substantial carbon reduction effects. The digital economy indirectly reduces CO_2_ emissions by expanding the economic scale of tertiary industries, reducing the proportion of coal consumption, and promoting green technology innovations [7]. However, since the application of information and communication technologies has increased energy consumption and pollution, the digital economy may also have negative externalities on the environment, resulting in an increase of carbon emissions. Therefore, it is necessary to investigate the relationship between the digitization of manufacturing investment and carbon emissions.

In this paper, the input digitalization of the manufacturing industry is defined as the value of the goods and services of all digital assets consumed and converted by all manufacturing industries in the process of producing or providing goods and services in a certain period (i.e., year). The input digitalization is classified into two types: productive input digitalization and distributive input digitalization. Productive input digitalization refers to the integration of digital technology and manufacturing technology. The digital input consumed in the production links the support of virtual reality, computer network, and other support technologies. Distributive input digitalization refers to the digitalized investment of the manufacturing industry in allocating resources, such as final goods or services, and the digitalized cost of goods in Internet retail. The key issue is the impact of manufacturing input digitization on the reduction of carbon emission intensity. In addition, there is a strong need to identify the types of input digitalization of the manufacturing industry that may have consistent impacts on carbon intensity, to find the transmission mechanisms of manufacturing service that affect carbon emission intensity, and to explore the characteristics of carbon emission reduction effects of input digitization of manufacturing in different industries and in different countries.

To answer these questions, we use WIOD input–output table and environmental ac-count data to calculate the input digitalization levels and carbon emission intensities of 17 manufacturing industries in China from 2000 to 2014. The fixed effect model is used to analyze the various impacts of productive input digitalization and distributive input digitalization on carbon emission intensity. This paper also carries out extensive analysis of the data, including the mechanism test of manufacturing input digitalization on carbon emission intensity, as well as the heterogeneity analysis of the manufacturing industry and the input digitalization sources.

The study of these issues will have not only substantial theoretical values, but also application significance for China and other developing countries to achieve both high-quality economic development and carbon emission reduction commitments. The potential contributions of this paper include the following aspects:(1)Based on the latest achievements of manufacturing input digitalization as the reference [8], we use the input–output method to calculate the input digitalization indicators. The complete carbon emission intensity index is used to express the carbon emission of the manufacturing industry for analyzing and accurately revealing the impact of input digitalization on carbon emission intensity.(2)We estimate the impact of different types of input digitization on carbon emission intensity and discuss the different roles of productive input digitalization and distributive input digitalization.(3)We include the cross-terms of input digitization and other variables into the model. The mechanisms of manufacturing input digitalization affecting carbon emission intensity are analyzed.(4)Multi-angle heterogeneity analysis is conducted, such as the impact of input digitization on the carbon emission intensity of different manufacturing industries and the impact of input digitization from different economic sources on carbon emission intensity. Finally, the conclusions and policy recommendations are targeted.

The rest of this paper is organized as follows. Section 2 reviews the literature related to this study, including the impact of digital industrialization on carbon emissions, the influence of manufacturing on carbon emissions, and the research progress for the input digitalization of manufacturing. Section 3 introduces the measurement model, model variables, and data sources. Section 4 shows the characteristics of China’s manufacturing input digitalization and carbon emission intensity from 2000 to 2014. Section 5 gives the empirical results and analysis. Section 6 provides conclusions and policy implications.

## 2. Literature Review

### 2.1. Impact of Digital Industrialization on Carbon Emissions

Digital industrialization is the process of digital upgrading, transformation, and reengineering of all elements in the upstream and downstream of the traditional industrial chain, empowered by the data. Research studies have been conducted in recent years regarding the environmental effects of industrial digitalization, but derived different conclusions.

Some studies propose that digital industrialization has positive impacts on carbon emissions. Industrial upgrading and technological progress have positive effects on stimulating the reduction of carbon emissions. For example, the application of wireless technology reduces the demand of physical office space for manufacturers. The popularity of a remote office mode has dramatically reduced the commuting demands related to work and shopping, which is conducive to decrease energy intensity [9]. The advanced communication technologies can reduce the energy intensity and pollutant emissions of enterprises, which has a role in promoting the quality of environment [10,11,12,13]. In addition, it is widely accepted that technological innovations have a key role in the control of environmental pollutions [14,15]. After digital factors become the new driving force, digitalization can significantly reduce the carbon emission intensity of Chinese manufacturing enterprises, and its impact is shown as a “marginal increase” [16]. Therefore, the digital development of manufacturing is the driving force of digital innovation that has positive impacts on carbon emission reduction.

However, digital industrialization may also have negative impacts on carbon emissions. The indirect energy plundering effect, caused by economic expansion and profit maximization, can be considered as a substitution effect. The integrated development of the digital economy and traditional industries will increase the revenues of production sectors. Driven by the increasing profits, enterprises tend to choose the continuous expansions of the production scale. Simultaneously, the scale of energy demand expands rapidly and the total carbon emissions also rise [17]. The negative impact of digital industrialization on energy consumption is not limited to the indirect energy plundering effect brought about by economic expansion. A large amount of energy consumption is needed to support the production, consumption, and disposal of the related products in energy-intensive industries [18]. Driven by profit maximization and constrained by bounded rationality, manufacturers will not produce products for the primary purpose of recycling and easy replacement, but increase the consumption of energy [19].

In summary, the impact of digital industrialization on carbon emissions mainly concentrates on the inhibition of income effects and the growth of substitution effects [20]. The mechanisms of income effects include the improvement of energy efficiency by reducing energy consumption, and optimization of industrial structure. The substitution effect refers to the energy consumption directly increased by the irrational application of digital technology, and the energy demand indirectly caused by the development of the digital economy. The appearance of the income effect often induces a substitution effect, and the improvement of the digital levels will also stimulate energy demand simultaneously [21]. The combination of income effects and substitution effects has a mixed impact on carbon emissions. Therefore, it is imperative to analyze the positive and negative effects of these factors.

### 2.2. Impact of Manufacturing on Carbon Emissions

The initial studies of carbon emissions from manufacturing focused on the perspective of international trade on the global carbon emissions, which gave rise to the “pollution paradise” hypothesis [22,23,24,25,26]. Using the input–output method [27], researchers started the studies for the carbon emissions embodied in China’s international trade. Research results show that the high growth of exports has rapidly driven the increase in China’s total carbon emissions [28,29,30,31].

In recent years there have been substantial research interests to analyze the environmental impact of manufacturing servitization by taking the subdivisions of the manufacturing industry as the research object. Relevant enterprise case studies show that service-oriented manufacturing enterprises can reduce resource consumption. Overall, this subdivision is beneficial to the sustainable development of the environment [32,33]. The input–output method is used to quantitatively measure the degree of input service in the manufacturing industry. Empirical tests have proved that the servitization of manufacturing can significantly reduce the level of carbon emissions in the industry [34,35].

The research on carbon emissions in China’s manufacturing industry mainly includes carbon emission decomposition and analysis of influencing factors. If we consider the total amount of carbon emissions only, the main factors affecting carbon emissions are energy intensity, energy structure, and economic growth [36]. However, if we consider economic growth primarily, efficiency improvement and structural optimization, driven by the progress of technologies, can effectively reduce carbon emissions. In addition, innovation is crucial to the green transformation and development of high-emission subsectors [37]. Therefore, the promotion of technological innovation and transformation, as well as the upgrade of the manufacturing industry, are conducive to the coordinated development of both the environment and economy [38,39,40].

### 2.3. Research on Digitalization of Manufacturing Investment

The development process of developed countries is generally followed by industrialization and then informatization. China needs to take into account informatization while developing industrialization. Today, the international division of labor is no longer limited to the manufacturing industry, but has rapidly expanded to the digital fields [41]. Digital manufacturing may be a possible approach to promote the transformation and upgrading of China’s manufacturing industry, and reshape the dynamics of global competition.

Under the framework of the division of the global labor and value chain, the relevant research studies concentrate on the levels of manufacturing enterprises and sectors. The results show that the level of input digitalization can affect the participation and status of the global value chains. On the enterprise level, scholars have theoretically discussed the mechanisms of digitalization in the upgrade of the enterprise value chain, reduction of transaction costs, and improvement of operational efficiency [42]. A few quantitative studies have analyzed the data at the province, industry, and enterprise levels separately. The results show that digitalization is an important approach for enterprises to enhance their participation in international trade and increase their trade income [43,44]. Manufacturers are exploring the extent to which digital technology applications can support their sustainability efforts to convert net-zero emissions and circular economy (CE) into feasible and practical actions, achievements, and ultimately, advantages of sustainable competition [45]. On the sector level, the study found that, through input digitalization to upgrade the global value chain of manufacturing industry and break through traditional trade, the main channels for value appreciation include the improvement of labor force levels, promotion of product upgrading, and generation of new production factors [46,47].

Although these theoretical and empirical studies have investigated the economic effects of manufacturing input digitalization, it is imperative to investigate the ecological and environmental benefits of manufacturing input digitalization. To address this issue, this study uses the input–output method to investigate the impact of input digitalization of manufacturing on carbon emissions and the corresponding transmission mechanisms.

## 3. Methodology, Variable Description and Data

### 3.1. Benchmark Regression Model

Based on the environmental pollution and supply model [48], we introduce digital manufacturing indicators and propose the following econometric model:(1)$$lnCarbon_{it}=\beta_{0\;}+\beta_{1}lnDigital_{it}+\beta_{2}X_{it}+\mu_{i}+\lambda_{t}+\varepsilon_{it},$$
where subscript i is the index of industry sector, t is time (year), $lnCarbon_{it}$ is the carbon emission intensity, independent variable $lnDigital_{it}$ refers to the digital level of the manufacturing industry, $\beta_{0\;}$, $\beta_{1}$, and $\beta_{2}$ are unknown coefficients to be estimated, $X_{it}$ is the control variable, $\mu_{i\;}$ and $\lambda_{t}$ represent the fixed effect of the industry and year, respectively, and $\varepsilon_{it}$ is the random error. To mitigate the adverse effects of heteroscedasticity, in addition to dummy variables, the variables in the model are treated with natural logarithms. Note that the majority of variable values are percentage data that are less than 1. To avoid the issue of heteroscedasticity caused by the negative values after taking logarithm, we add 1 to the original value and then take the logarithm.

### 3.2. Variable Description

#### 3.2.1. Carbon Emission Intensity (Carbon)

According to the measurement method for China’s regional carbon emission intensity [49], the total carbon emission intensity of the manufacturing industry calculated by the input–output model is given by:(2)$$\left[Carbon\right]=\left[\frac{\;C_{1}}{A_{1}}\;\frac{\;C_{2}}{A_{2}}\;\cdots \frac{\;C_{j}}{A_{j}}\;\cdots \;\frac{\;C_{n}}{A_{n}}\right]\left[{\left(I-A\right)}^{-1}-I\right],$$
where j is the index of the manufacturing sector, $C_{j}$ is the direct carbon emissions of sector j. $A_{j}$ is the total output of sector j, I is the identity matrix, $\left[{\left(I-A\right)}^{-1}-I\right]$ is the inverse matrix of Leontief, and Carbon is the row vector of the total carbon emission intensity of manufacturing.

#### 3.2.2. Input Digitalization Level (Digital)

In this paper, the digitalization of manufacturing is defined as the degree of intermediate digitalization input in the manufacturing industry, which comes from the industries containing digital economic elements. Based on the International Standard Industrial Classification (ISIC Rev4.0) of the United Nations Statistics Agency, we use the digital economy elements of the manufacturing industry to define the digital related industries of intermediate input sources (see Table 1 for details). The J58, G46 and G47 industries only include part of the digitalization contents. Thus, the digitalization departments are divided into J58-d, G46-d, and G47-d based on the proportions of digitalization.

We use the direct consumption coefficient and indirect consumption coefficient to measure [*Digital*] in the input–output method [50,51]. The direct consumption coefficient of input digitalization reflects the direct consumption of digital intermediate products in the production process of department unit products, given by:(3)$$d_{ij}=\frac{a_{ij}}{A_{j}},\;i=1,2,\cdots ,m,j=1,2,\cdots ,n,$$
where j is the index of the manufacturing sector, and i represents the sector providing digital intermediate products, $d_{ij}$ is the consumption of digital products of department i per unit product produced by department j, $a_{ij}$ is the number of digital products of the ith department consumed by the j department, and $A_{j}$ is the total output of sector j.

The total consumption coefficient measures the level of input digitalization by using the sum of direct and indirect consumptions of digital products by the manufacturing industry, given by:(4)$$D_{ij}=d_{ij}+\sum_{k=1}^{n}d_{ik}d_{kj}+\sum_{s=1}^{n}\sum_{k=1}^{n}d_{is}d_{sk}d_{kj}+\cdots ,i=1,2,\cdots ,m,j,k,s=1,2,\cdots ,n,$$
where $D_{ij}$ is the total consumption coefficient of the unit product of department j to the digital product of department $i,\;d_{ij}$ is the direct consumption of digital products of department i by unit products produced by department j, $\sum_{k=1}^{n}d_{ik}d_{kj}$ is the first indirect consumption of unit products produced by department j, and $\sum_{s=1}^{n}\sum_{k=1}^{n}d_{is}d_{sk}d_{kj}$ is the second indirect consumption. The digitalization level of manufacturing sector j is defined by:(5)$$Digital_{j}=\sum_{i=1}^{m}D_{ij},i=1,2,\cdots ,m,j=1,2,\cdots ,n.$$

#### 3.2.3. Production Input Digitalization Level (PRODIG) and Distributive Input Digitalization Level (DISDIG)

The service industry system in Singman’s service industry quartering method has been simplified and classified as the productive service industry, distributive service industry, consumer service industry, and social service industry. Since the majority of industries that input digitalization is invested in belong to the service industry, we classify input digitalization into three major types: productive input digitalization, distributive input digitalization, and consumer input digitalization. Based on this classification, we can discuss the heterogeneity of the impact of digital input on carbon emission intensity. It should be noted that there is no input data in WIOD input–output data for the J59-J60 industry in China from 2000 to 2014. Therefore, the following analysis focuses on the impact of productive and distributive digitalization on carbon emissions. The digitalization level of productive input and distributive input of manufacturing sector j are:(6)$$PRODIG=\sum_{i=1}^{m_{p}}D_{ij},DISDIG=\sum_{i=1}^{m_{d}}D_{ij},\;m_{p}+m_{d}=m.$$
where $m_{p}$ is the number of departments with productive input digitalization, $m_{d}$ is the number of departments with distributive input digitalization.

#### 3.2.4. Control Variable

Energy consumption intensity (*ECI*) is defined by the ratio of energy consumption of each manufacturing industry to the output value of this industry.

Energy consumption structure (*ECS*) is the ratio of non-fossil energy consumption of each manufacturing industry to the energy consumption of this industry.

Industry scale (*INDS*) is expressed by the ratio of the output value of each industry to the total output value of the manufacturing industry. It is used to control the scale effect of the manufacturing industry.

Capital labor ratio (*CLR*) gives the ratio of the total value of fixed assets of each manufacturing industry to the total number of employees in this industry. It is used to control the structural effect of the manufacturing industry.

R&D expenditure (*RDS*) is the proportion of the internal R&D expenditure of each manufacturing industry in the total output value of this industry. It is used to control the technological effect of the manufacturing industry.

Foreign direct investment (*FDI*) is expressed by the proportion of the total industrial output value of foreign investment and Hong Kong, Macao, and Taiwan investment in the total output value of this industry. It is used controls the scale, structure, and technology effects of FDI spillovers to host countries.

Environmental regulation (*ERI*) are also important, as enterprises are likely to increase pollution control expenditure and reduce pollution emissions due to environmental regulations issued by the government. This variable controls the binding effect of social responsibility on the manufacturing industry. There are three mainstream accounting indicators of environmental regulation: pollution discharge cost, pollution emission intensity, and the ERS comprehensive index [52,53]. This paper mainly refers to the third method and establishes an indicator system from positive and negative aspects to comprehensively measure the intensity of environmental regulation, as shown in Table 2.

### 3.3. Data Source

The original data for measuring the digital level of manufacturing inputs come from the input–output table of WIOD (2016), which provides the inter country input and output data of 56 industries between 44 major countries from 2000 to 2014. The data required for carbon emission intensity calculation are from the WIOD environmental account data. The original data used for control variables are from the *China Statistical Yearbook*, *China Industrial Statistical Yearbook*, *China Science and Technology Statistical Yearbook*, and *China Environmental Statistical Yearbook*. Since WIOD (2016) adopts the International Standard Industry Classification (ISIC Rev. 4), while China’s manufacturing industry classification is based on the National Economic Standard Industry Classification, these two classification systems are different. Therefore, we integrate the research objects into 17 manufacturing industries according to the two systems, as shown in Table 3.

All data related to the gross industrial output value, fixed assets and other currencies are deflated using the relevant price index with the base period of 2000, to obtain constant price data. Some missing data are processed by interpolation or moving average. The sample data cover 17 manufacturing sectors in China from 2000 to 2014. The statistical description of the variables explained in Section 3.2 are shown in Table 4.

## 4. Characteristics of Input Digitalization and Carbon Emission Intensity

### 4.1. Distribution of Manufacturing Input Digitalization and Carbon Emission Intensity

We first qualitatively analyze the characteristics of manufacturing input digitalization and carbon emission intensity. Figure 1 shows the digital ridge map of input and carbon emission intensity of 17 manufacturing industries from 2000 to 2014. There is a significant difference between the industries in the digitalization investment. The manufacturing industries I1–I13 have a relatively concentrated input digitalization and a similar distribution pattern, with only one peak and even distribution. The distributions of industries I14–I17 are somewhat scattered and the distribution patterns are different. Generally, industries with higher technology application levels have higher input digitalization levels. Figure 1b shows that the distribution patterns of carbon emission intensity of manufacturing industries are similar and have two peaks. The distribution of carbon emission intensity in each industry is relatively dispersed, which is basically consistent with the expected carbon emission caused by the nature of the industry. For example, industries I10, I11, and I12 (i.e., manufacturing of other non-metallic mineral products, primary metals, and metal products) have almost identical distributions and have large values of carbon emission intensities.

### 4.2. Average and Development of Manufacturing Input Digitalization and Carbon Emission Intensity

Figure 2a shows the average digital levels of manufacturing investment from 2000 to 2014. The degree of digitalization of China’s manufacturing investment is relatively low. In particular, the distributive input digitalization is below 2% during the majority of this time period. The changing trend of productive input digitalization is almost identical to that of the overall digitalization investment, and the investment level has been increased by about 50% in the past 15 years. The fluctuation trend of distributive input digitization is different, and the fluctuation range is small. Generally, the dynamic evolutionary process of manufacturing input digitalization can be divided into two stages. The first stage is from 2000 to 2008. The levels of input digitalization gradually declined after reaching the peak in 2005, and reached the bottom in 2008. The second phase is from 2009 to 2014, in which the input digitalization levels continued to grow. Figure 2b–d, respectively, gives the development of input digitalization, productive input digitalization and distributive input digitalization of 17 manufacturing industries from 2000 to 2014. The changing trends of input digitalization and productive input digitalization in various manufacturing industries are relatively consistent to each other. However, the changing trend of distributive input digitalization is different from the other two trends.

Figure 3a shows the average carbon emission intensity of the manufacturing industry from 2000 to 2014. The average carbon emission intensity of the manufacturing industry shows a downward trend, indicating that China’s manufacturing industry has achieved remarkable green governance results. Note that 2008–2009 is the only time period showing an increase of the average carbon emission intensity. This increase may be due to the impact of the financial crisis in this time period. Figure 3b shows the development of carbon emission intensity of 17 manufacturing industries from 2000 to 2014. The carbon emission intensity of each manufacturing industry has shown a downward trend over time.

## 5. Results

### 5.1. Benchmark Regression

The benchmark regression results in Table 5 show two cases when only core explanatory variables and control variables are considered. The core explanatory variables passed the statistical test at the significance level of 1%.

Columns (1) and (2) in Table 5 show that the coefficient of *lnDigital* is significantly negative, indicating that the input digitalization of manufacturing has significantly reduced the carbon emission intensity. Columns (3)–(8) consider different types of input digitalization. Columns (3) and (4) show that the coefficient of *lnPRODIG* is negative, indicating that productive input digitalization is conducive to reducing carbon emission intensity. Interestingly, columns (5) and (6) show that the coefficient of *lnDISDIG* is significantly positive, suggesting that the distributive input digitalization may aggravate the carbon emission. The reason may be that the digitalization of distribution is difficult to replace the traditional labor and material resources, and the development of this new model requires more energy consumption. The results of digital segmentation show that different types of input digitalization have different impacts on the carbon emission intensity. For comparative analysis, columns (7) and (8) include *lnPRODIG* and *lnDISDIG* in the measurement model, respectively. The results show that the digitalization of productive input and digitalization of distributive input have opposite effects on the carbon emission intensity of the manufacturing industry.

The application cost of productive digital technology is low, which positively affects carbon emission reduction in the manufacturing industry, significantly. However, the transformation of distributive input digitalization, such as internet trade and online retail, has high transport costs and is easily restricted by digital trade barriers, which leads to an increase in the carbon emission intensity. Since distributive digitalization accounts for a low proportion of overall input digitalization, the comprehensive impact of input digitalization on reducing carbon emission intensity is still positive. The estimated results of control variables through statistical tests are consistent with expectations. However, the effects of some control variables on carbon emission intensity are not robust. The reason may be that the two types of input digitization have different impacts on carbon emission intensity.

### 5.2. Robustness Test

#### 5.2.1. Change the Sample Period

Since advanced countries were greatly affected by the 2008 financial crisis, their input digitalization investment in China’s manufacturing industry were changed substantially. The trend of China’s manufacturing carbon emission intensity has also changed. Therefore, two sample time periods, namely 2000–2008 and 2008–2014, are selected to estimate the impact coefficient of input digitalization on carbon emission intensity, to verify the robustness of the model conclusions. Columns (1)–(4) in Table 6 are the baseline regression results for the time period of 2000–2008. The results show that the coefficients of input digitalization and productive input digitalization are significantly negative, while the coefficients of distributive input digitalization are significantly positive. Columns (5)–(8) are the benchmark regression results for the time period 2009–2014. The results show that the coefficient signs and significance of *lnDigital*, *lnPRODIG*, and *lnDISDIG* are consistent with those of 2000–2008. Compared with the results in Table 5, the symbol and significance of the core variable coefficients of the model have not changed after the sample time period is separated, and the model results are relatively stable.

#### 5.2.2. Endogenous Test

Manufacturing industries with a lower carbon emission intensity may be more technologically mature and have low production cost for digitalization, making it easier to promote industrial digital transformation. Therefore, there may be endogenous problems caused by the reverse causality between input digitization and carbon emission intensity. The input digitization level *L.lnDigital*, lagged by one period (productive digital input *L.lnPRODIG* lagged one period, distribution digital input *L.lnDISDIG* lagged one period), is used as the instrumental variable of manufacturing input digitization lnDigital (productive digital input *lnPRODIG*, distribution digital input *lnDISDIG)*. Then we estimate the impact coefficients on carbon emission intensity by a two-stage least square method (*2SLS*).

Table 7 shows the results of the endogenous test. The columns (1), (2), and (3) show that Kleibergen-Paap rk Wald F has a threshold of 16.38 at the 10% significant level, all of which significantly reject the null hypothesis of “weak recognition of instrumental variables” at higher levels. Kleibergen-Paap rk LM has a *p*-value of 0, which significantly rejects the null hypothesis of “insufficient recognition of tool variables” at higher levels. The results suggest that the estimation results all significantly reject the null hypothesis of insufficient identification and weak instrumental variables. The coefficients of input digitization and productive input digitization are significantly negative, and the coefficients of distributive input digitization are significantly positive. After considering possible endogenous problems, the core conclusion of this paper is still robust.

### 5.3. Mechanism Test

Next, we discuss the mechanisms by which manufacturing invests in digitization to reduce carbon emission intensity. Referring to the previous analysis, the following approaches may help reduce carbon emission intensity:(1)Scale effect: increase the industry scale of the manufacturing industry and absorb more digital elements.(2)Structural effects: improve the substitution of capital (including digital capital) for labor factors and optimize the industrial structure.(3)Technological effects: increase manufacturing R&D spending and upgrade low-carbon technology levels.(4)Spillover effects: increase the proportion of foreign direct investment and increase the technological and human spillovers of multinational enterprises to the host country (China).(5)Binding effects: strengthen the monitoring of environmental regulation and strictly restrain the negative impact of the manufacturing industry on the environment.

To verify the above effects, we add manufacturing industry scale (*lnINDS*), capital-labor ratio (*lnCLR*), R&D expenditure (*lnRDS*), foreign direct investment (*lnFDI*), environmental regulation (*lnERI*) and core variables (*lnDigital*) in the benchmark regression model, to examine the path of input digitalization to reduce carbon emission intensity. The principle of this test is that if the sign of the test result for lnDigital is negative and the sign of the cross item is positive, it means that this effect mechanism has a more significant role in reducing carbon emission intensity.

Table 8 shows the results of the mechanism test. Column (1) gives test results for the scale effect. The coefficient of *lnDigital* is −0.378, which means that for every one percentage increase in the digitalization of manufacturing input, the carbon emission intensity will decrease by 0.378 percentages. The effect of emission reduction is remarkable. The coefficient of the cross-term *lnINDS* × *lnDigital* is significantly negative, which indicates that expanding the scale of the industry can amplify the role of input digitalization in reducing carbon emission intensity. Column (2) is a mechanism test for structural effects. The coefficients of *lnDigital* and the cross-term *lnCLR* × *lnDigital* are significantly negative, indicating that the higher the proportion of capital factors, the more important the role of input digitalization in reducing carbon emission intensity. Thus, the results verify the structural optimization path for the effect of investment in digitalization to reduce carbon emission intensity.

Column (3) shows the mechanism test results for technology effects. The coefficient of *lnDigital* is significantly negative, and the coefficient of the cross-term *lnCLR* × *lnDigital* is negative but not significant, which is inconsistent with the expected results. The possible reason may be that the current industrial digitalization is still in the immature stage of rapid development. R&D investment in new digital technologies has increased significantly, which requires more production resources and energy input, thus triggering an increase in carbon emissions. However, as the application scale of newly developed digital technologies continues to expand and the degree of application continues to deepen, more and more traditional production factors will be replaced by digital factors, playing a positive role in reducing carbon emission intensity. The offset of the above two effects may suggest that the mechanism of input digitalization in reducing carbon emission intensity through technical effects is not significant. Columns (4) and (5) are the mechanism test results for the spillover effect of foreign direct investment and the restrictive effect of environmental regulation. The coefficients of *lnDigital* and the cross-terms *lnFDI* × *lnDigital* and *lnERI* × *lnDigital* all are significantly negative, indicating that digital investment can be achieved by increasing direct foreign investment, or that enhanced environmental regulation can promote carbon emission reduction.

### 5.4. Heterogeneity Analysis

Due to the difference in product types, development models and technical levels among manufacturing industries and the degree of environmental pollution are also different. Therefore, input digitalization may have a heterogeneous impact on the carbon emission intensity of different manufacturing industries. This subsection will explore this heterogeneity.

#### 5.4.1. Pollution Level of the Manufacturing Industry

The manufacturing industry is separated into two types: pollution-intensive manufacturing and non-pollution-intensive manufacturing, according to the degree of industrial pollution (pollution-intensive manufacturing industries include I1, I2, I3, I5, I6, I8, I9, I13, I14, I15, I16 and I17; non-pollution-intensive manufacturing industries include I4, I7, I10, I1 and I12). We set a dummy variable *Clean*, where *Clean* = 1 represents the non-pollution-intensive manufacturing industry, and the pollution-intensive manufacturing industry (*Clean* = 0) is the benchmark group. We introduce the cross-item *Clean* × *lnDigital* (*Clean* × *lnPRODIG*, *Clean* × *lnDISDIG*) to analyze the differential impact of input digitalization (productive input digitalization and distributive input digitalization) on the carbon emission intensity of manufacturing industries with different pollution intensities. The results are shown in columns (1)–(3) of Table 9.

Column (1) of Table 9 shows that the coefficient of *lnDigital* is significantly negative, indicating that input digitalization can reduce the carbon emission intensity of pollution-intensive manufacturing. The coefficient of *Clean* × *lnDigital* is significantly negative, meaning that the input digitalization of the non-pollution-intensive manufacturing industry has a more substantial inhibitory effect on carbon emission intensity. For every 1% increase in input digitalization, the carbon emission intensity of pollution-intensive and non-pollution-intensive manufacturing will decrease by 0.390% and 0.550%, respectively.

Column (2) shows that the coefficient of *lnPRODIG* is also significantly negative, indicating that productive input digitalization can reduce the carbon emission intensity of pollution-intensive manufacturing. The coefficient of the cross-item *Clean* × *lnPRODIG* is positive but not significant, indicating that the productive input digitalization in the two manufacturing industries has no significant difference in the inhibition of carbon emission intensity.

However, column (3) shows that the coefficient of *lnDISDIG* is significantly positive, indicating that distributive input digitalization will increase the carbon emission intensity of the pollution-intensive manufacturing industry. The coefficient of *Clean* × *lnDISDIG* is significantly positive, indicating that the distributive input digitalization in non-pollution-intensive manufacturing increases carbon emission intensity even more than pollution-intensive manufacturing. The carbon emission intensity of the pollution-intensive manufacturing industry and non-pollution-intensive manufacturing industry will increase by 6.510% and 8.417%, respectively, for each 1% increase in the distributive input digitalization.

#### 5.4.2. Input Digitalization Level of the Manufacturing Industry

According to the median of the input digitalization level of each manufacturing industry in 2014, we classify the manufacturing industries that are above this median into high-digital-input manufacturing industries, and low-input digital manufacturing for the rest of the manufacturing industry (high-digital-input manufacturing industries include I5, I7, I12, I13, I14, I15, I16 and I17; low-digital-input manufacturing industries include I1, I2, I3, I4, I6, I8, I9, I10 and I11). We set a dummy variable *High*, where *High* = 1 represents the high-digital-input manufacturing industry, and *High* = 0, the low-digital-input manufacturing industry, as the benchmark group. The cross-term *High* × *lnDigital* (*High* × *lnPRODIG*, *High* × *lnDISDIG*) is introduced to investigate the differential impact of input digitalization (productive input digitalization, distributive input digitalization) on the carbon emission intensity of the manufacturing industry at different input digitalization levels. The results are shown in columns (4)–(6) of Table 9.

Column (4) of Table 9 shows that the coefficient of *lnDigital* is significantly negative, and the coefficient of the cross-term *High* × *lnDigital* is also significantly negative, indicating that the input digitalization of the high-digital-input manufacturing industry has a more substantial inhibitory effect on carbon emission intensity. For every 1% increase in input digitalization, the carbon emission intensity of the high-digital-input and low-input digital manufacturing industries will decrease by 0.540% and 0.389%, respectively.

Column (5) shows that the coefficients of *lnPRODIG* and *High* × *lnPRODIG* are significantly negative, indicating that the productive input digitalization in these two types of manufacturing industries has a significant difference in the suppression of carbon emission intensity. The productive input digitalization in the high-digital-input manufacturing has a more substantial inhibitory effect on carbon emission intensity than the low-digital-input manufacturing. The carbon emission intensity of the high-digital-input manufacturing industry and low-digital-input manufacturing industry will decrease by 0.952% and 0.790%, respectively, for every 1% increase in productive input digitalization.

However, column (6) shows that the coefficients of *lnDISDIG* and the cross-item *High* × *lnDISDIG* are significantly positive, indicating that distributive input digitalization will increase the carbon emission intensity of high-digital-input manufacturing. The carbon emission intensities of low-digital-input and high-input digital manufacturing will increase by 5.414% and 7.745%, respectively, for each 1% increase in distributive input digitalization. The possible reasons may be that China’s manufacturing industry, which has a high degree of digital investment, is currently at a critical stage of tackling digital transformation and distribution, together with the existence of digital trade barriers, and the lack of ability to reduce energy consumption per unit of product.

### 5.5. Complementary Analysis

Considering the absorption and transformation of digital elements from domestic and foreign sources in the manufacturing industry, the impact of input digitalization on carbon emission intensity may be different.

We divide the multi-regional input–output table provided by WIOD into two parts: China’s own input from domestic sources and input from other countries. Then we calculate the domestic input digitalization rate (*DDIG*) and foreign input digitalization rate (*FDIG*) of manufacturing. Figure 4 shows the development trend and composition of input digitalization from domestic and foreign sources in 2000–2014. It can be seen that the digitalization of China’s manufacturing industry mainly comes from domestic investment, and the input rate from foreign countries is low. The input digitization from the two sources has different trends over time. The overall domestic input digitalization is on the rise, especially since the financial crisis in 2008. Foreign input digitalization has had a growth trend since 2000, peaked in 2006, and then declined. China’s manufacturing industry has moved towards independent digitalization in recent years, and its dependence on foreign digital elements has decreased.

We add *lnDDIG* and *lnFDIG* into the econometric model to explore the impact of national source heterogeneity on carbon emission intensity. According to Table 10, the core explanatory variables from both sources passed the statistical test at the 1% significance level. Column (1) indicates that the coefficient of input digitalization from China is significantly positive. That is, the domestic input digitalization helps reduce the carbon emission intensity of the manufacturing industry. Column (2) indicates that input digitalization from other countries fails to reduce the carbon emission intensity of China’s manufacturing industry, but increases energy consumption costs.

Column (3) includes *lnDDIG* and *lnFDIG* into the model and compares them with the benchmark results in columns (1) and (2). The absolute values of the estimated coefficients of *lnDDIG* and *lnFDIG* decrease significantly. Every 1% increase in domestic input digitalization will reduce the carbon emission intensity by 0.745%, but every 1% increase in foreign input digitalization will increase the carbon emission intensity by 3.775%. The possible reasons for this heterogeneous result may include the following two aspects: on the one hand, China has an enormous potential for carbon emission reduction, and the digital economy is developing rapidly. The domestic digital elements are abundant and easy to obtain, and manufacturing is in a rapid digitalization process. The reform combined with energy structure, low-carbon technology, and other aspects can significantly promote manufacturing carbon emission reduction. On the other hand, introducing digital technology from other economies requires more cross-border transport costs. Due to the digital trade barriers, China’s ability to trigger a change is relatively limited. It is not as good as the domestic input digitalization, that can bring obvious carbon emission reduction dividends. It may even hinder the process of carbon emission reduction in the manufacturing industry to a certain extent.

## 6. Conclusions

To explore the mechanisms for developing the manufacturing industry and maintaining ecological environment, this paper focuses on the emerging industrial development trend of digital investment. From the perspective of input and output, we analyze the digital levels of input and carbon emission intensities of 17 manufacturing industries in China. This empirical study examines the impact of the digital levels of manufacturing investment on the carbon emission intensity. We also explore the mechanisms regarding the influence of input digitalization on the carbon emissions of the manufacturing industry. In addition, we analyze the industry heterogeneity in the digital carbon emission reduction mechanism and the heterogeneity of the impact of input digitization from different countries on carbon emissions. The major conclusions of this study are that:(1)The digital level of China’s manufacturing investment is on the rise, while the carbon emission intensity declines simultaneously. Both were affected by the financial crisis in 2008, and the trend variated. From the attribute of input digitalization, the proportion of productive input digitalization in the manufacturing industry is relatively high. In contrast, the proportion of distributive input digitalization is relatively low.(2)The improvement of manufacturing input digitalization can have a significant positive impact on carbon emission reduction. Among them, the productive input digitalization has a positive role in promoting carbon emission reduction in manufacturing. In contrast, the distributive input digitalization has a negative role in inhibiting carbon emission reduction in manufacturing. This conclusion is robust after changing the sample time period and dealing with sample endogeneity (i.e., using instrumental variable estimation).(3)The mechanism analysis results show that the input digitalization promotes the carbon emission reduction process by exerting the scale, structure, spillover, and constraint effects. In contrast, the emission reduction effect is not apparent through new digital technology.(4)The industry heterogeneity analysis shows that input digitalization plays a significant role in reducing the carbon emission intensity of non-pollution-intensive manufacturing. The input digitalization of manufacturing with a higher degree of digitalization has a corresponding stronger inhibition on carbon emission intensity.(5)The analysis of national sources of input digitalization shows that input digitalization relying on domestic sources has a positive role on carbon emission reduction in manufacturing, while input digitalization relying on foreign sources has a negative role in inhibiting carbon emission reduction in the manufacturing industry.

The above conclusions confirm the positive effect of input digitalization on carbon emission reduction. However, two different types of input digitalization have an opposite impact on carbon emissions. The impact of each manufacturing industry on its total carbon emission depends on the balance of these two opposite forces. Since the technology effect of input digitalization is mainly dependent on distributive digitalization, the final comprehensive impact on carbon emission intensity is not significant. Although the impact of distributive input digitalization on inhibiting carbon emission intensity is negative, the conclusions of this study still have important application significance. The reason is that, if we do not realize this problem, the manufacturing industry that blindly relies on distributive digitalization may ignore this issue, which leads to the painful reality that the total carbon emissions will continue to grow.

Based on the above research conclusions, we put forward the following policy recommendations:(1)The government should fully seize the opportunity of digital development, break through the bottleneck of core digital technology, cultivate comprehensive digital talents, and strengthen the ability of the manufacturing industry to acquire and apply digital elements.(2)The government should pay attention to the heterogeneity of the impact of input digitalization from different manufacturing sectors on carbon emissions. Different levels of carbon emission reduction targets should be formulated according to the development characteristics of each manufacturing industry. Non-pollution-intensive manufacturing or industries with a high degree of digitalization can achieve more significant environmental effects through digitalization.(3)Manufacturing industries should strengthen independent research studies and development of digital technology. Enterprises should absorb and integrate digital elements according to the industry characteristics to achieve the win-win goal for both the structural transformation of the manufacturing industry, and carbon emission reduction.

In the context of the international division of labor and global carbon emission reduction, our study provides quantitative evidence of the impact of digital input on the ecological and environmental benefits of the manufacturing industry. This paper enriches and expands the economic and ecological benefits of digitalization from theoretical and empirical aspects. Finally, it is worth noting that although this work aimed to study the carbon emission reduction of China’s manufacturing industry, the proposed research principles and methodologies can be applied to the study of the impact of input digitization on carbon emission intensity in other countries.

Although we have analyzed the impact of input digitalization on carbon emission reduction, there are still many important questions requiring further analysis. In this work we study 17 manufacturing industries in China. However, the enterprise levels are not included in the research framework. It is expected that the study results may show different influences of enterprise levels on carbon emission. Thus, we need to investigate the micro-effects of enterprise levels. In addition, our analysis only draws the conclusion that the digitalization of foreign investment harms carbon emission reduction. We did not discuss the input digitalization heterogeneity from different countries with different levels of national digital economy development, due to the lack of statistical data and measurement indicators. Currently, there is not a uniform measurement method for digital development levels in different countries. The relative levels of input digitalization relying on foreign sources need to be precisely rescaled. Therefore, it is imperative to conduct micro-effect analysis, depict the degree of input digitalization of various countries accurately, and integrate multiple world economies into a uniform analysis framework to make broad and deep conclusions. All these issues will be topics of future research.

## Figures and Tables

**Figure 1 ijerph-20-03174-f001:**
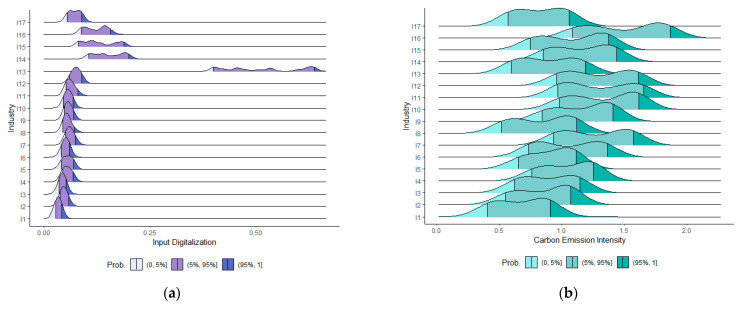
(**a**) Distributions of input digitalization. (**b**) Distributions of carbon emission intensity.

**Figure 2 ijerph-20-03174-f002:**
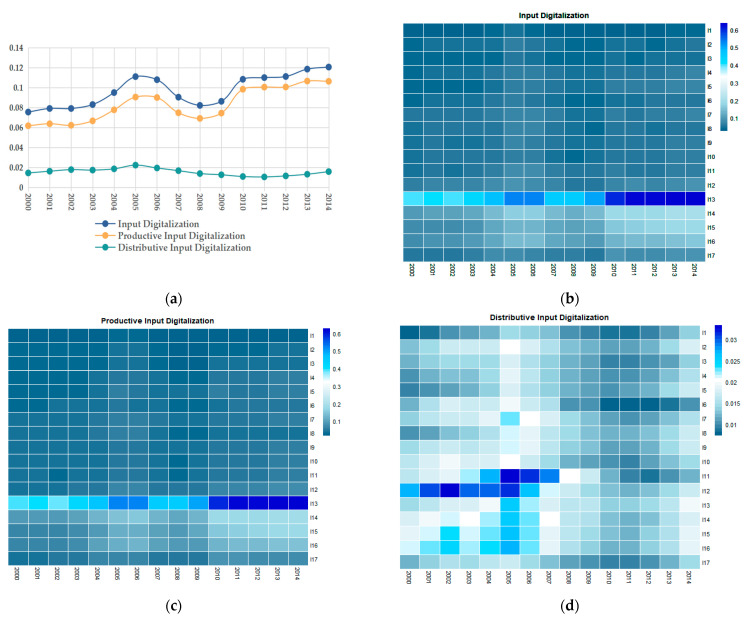
(**a**) Average input digitalization of 17 manufacturing industries from 2000 to 2014; (**b**) Development of input digitalization in 17 manufacturing industries from 2000 to 2014; (**c**) Development of productive input digitalization in 17 manufacturing industries from 2000 to 2014; (**d**) Development of distributive input digitalization in 17 manufacturing industries from 2000 to 2014.

**Figure 3 ijerph-20-03174-f003:**
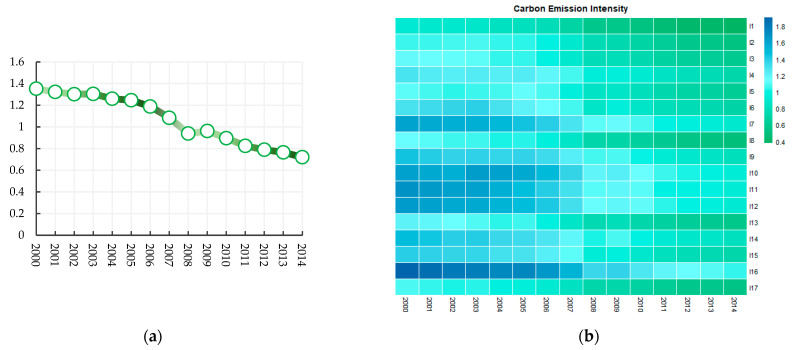
(**a**) Average carbon emission intensity of 17 manufacturing industries from 2000 to 2014; (**b**) Development of carbon emission intensity in 17 manufacturing industries from 2000 to 2014.

**Figure 4 ijerph-20-03174-f004:**
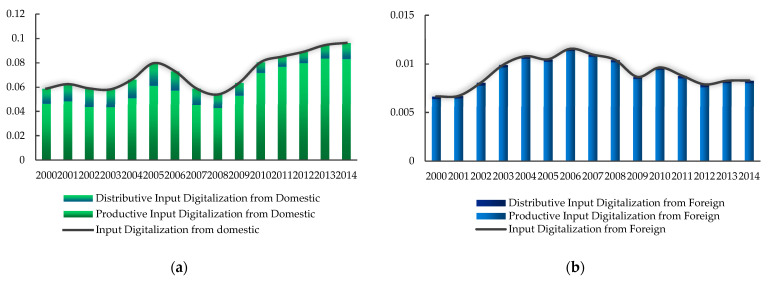
(**a**) Development and composition of input digitalization from domestic sources; (**b**) Development and composition of input digitalization from foreign countries.

**Table 1 ijerph-20-03174-t001:** Classifications of digital industry.

	Industry Classification (ISIC Rev4.0)	Industry Description	Digital Industry Classification
1	C26	Manufacture of computers, electronic and optical products	Productive
2	J58-d:5819;5820	Other publishing activities and software distribution	
3	J62-J63	Computer programming, consulting and related activities and information service activities	
4	G46-d	Internet wholesale and trade agents	Distributive
5	G47-d	Internet retail	
6	J61	Telecommunication	
7	J59-J60	Film, video and television program production, audio recording and music work publishing activities, radio and program production activities	Consumption

**Table 2 ijerph-20-03174-t002:** The measurement system of environmental regulations (+: positive influence, −: negative influence).

Evaluation Objects	Selected Indicators	Indicator Representation	Direction of Influence
Environmental regulation process	Wastewater treatment rate	Industrial COD removal rate	+
	Waste gas treatment rate	Industrial SO_2_ removal rate	+
	Solid waste treatment rate	Comprehensive utilization rate of industrial solid waste	+
Effectiveness of environmental regulation	Effluent discharge intensity	COD discharged per unit industrial output value	−
	Exhaust emission intensity	SO_2_ emission per unit industrial output value	−
	Discharge intensity of solid waste	Solid wastes discharged per unit industrial output value	−

**Table 3 ijerph-20-03174-t003:** List of 17 manufacturing industries.

Manufacturing Industry	ISIC Rev4.0 Code	ISIC. 4 Industry Definition
I1	C10–C12	Manufacture of food products, beverages and tobacco products
I2	C13–C15	Manufacture of textiles, wearing apparel and leather products
I3	C16	Manufacture of wood and of products of wood and cork, except furniture; Manufacture of articles of straw and plaiting materials
I4	C17	Manufacture of paper and paper products
I5	C18	Printing and reproduction of recorded media
I6	C19	Manufacture of coke and refined petroleum products
I7	C20	Manufacture of chemicals and chemical products
I8	C21	Manufacture of basic pharmaceutical products and pharmaceutical preparations
I9	C22	Manufacture of rubber and plastic products
I10	C23	Manufacture of other non-metallic mineral products
I11	C24	Manufacture of basic metals
I12	C25	Manufacture of fabricated metal products, except machinery and equipment
I13	C26	Manufacture of computer, electronic and optical products
I14	C27	Manufacture of electrical equipment
I15	C28	Manufacture of machinery and equipment n.e.c.
I16	C29–C30	Manufacture of motor vehicles, trailers and semi-trailers; Manufacture of other transport equipment
I17	C31–C32	Manufacture of furniture; Other manufacturing

**Table 4 ijerph-20-03174-t004:** Descriptive statistics of 11 variables.

Variables	Obs.	Mean	Std. Dev.	Min.	Max.
*lnCarbon*	255	0.3659	0.1478	0.0987	0.8241
*lnDigital*	255	0.0972	0.1124	0.0270	0.6384
*lnPRODIG*	255	0.0829	0.1131	0.0176	0.6289
*lnDISDIG*	255	0.0155	0.0050	0.0073	0.0334
*lnECI*	255	1.5556	0.9526	0.2620	4.4993
*lnECS*	255	1.5893	1.1660	0.2956	3.7361
*lnINDS*	255	9.3852	1.1355	6.4260	11.4491
*lnCLR*	255	2.5693	0.5919	1.3884	4.3150
*lnRDS*	255	0.4631	0.2780	0.0565	1.1207
*lnFDI*	255	3.4408	0.4958	1.7873	4.6321
*lnERI*	255	0.6426	0.0776	0.3400	0.7439

**Table 5 ijerph-20-03174-t005:** Benchmark regression results of model (1), with core explanatory variables and control variables.

Variable	Dependent Variable: *lnCarbon*							
	(1)	(2)	(3)	(4)	(5)	(6)	(7)	(8)
*lnDigital*	−0.480 ***	−0.514 ***						
	(−3.02)	(−3.53)						
*lnPRODIG*			−1.021 ***	−0.932 ***			−0.489 ***	−0.499 ***
			(−5.57)	(−5.61)			(−2.94)	(−3.27)
*lnDISDIG*					8.467 ***	7.115 ***	7.632 ***	6.317 ***
					(10.77)	(10.02)	(9.27)	(8.59)
*lnECI*		0.059 ***		0.059 ***		0.050 ***		0.052 ***
		(4.55)		(4.68)		(4.55)		(4.80)
*lnECS*		0.036 **		0.044 ***		0.046 ***		0.051 ***
		(2.26)		(2.82)		(3.38)		(3.82)
*lnINDS*		−0.002		−0.002		−0.002		−0.002
		(−1.21)		(−0.93)		(−1.57)		(−1.55)
*lnCLR*		−0.033 **		−0.023		−0.028 **		−0.016
		(−2.00)		(−1.41)		(−2.00)		(−1.13)
*lnRDS*		−0.012		−0.011		−0.006		−0.006
		(−1.02)		(−0.97)		(−0.56)		(−0.60)
*lnFDI*		−0.069 ***		−0.067 ***		−0.067 ***		−0.064 ***
		(−6.09)		(−6.15)		(−7.06)		(−6.84)
*lnERI*		−0.112 *		−0.122 *		0.015		−0.039
		(−1.67)		(−1.94)		(0.27)		(−0.71)
Constant	0.540 ***	0.746 ***	0.552 ***	0.711 ***	0.380 ***	0.522 ***	0.415 ***	0.539 ***
	(38.54)	(10.67)	(49.89)	(10.63)	(29.48)	(8.37)	(23.72)	(8.79)
Industry FE	YES	YES	YES	YES	YES	YES	YES	YES
Year FE	YES	YES	YES	YES	YES	YES	YES	YES
Observations	255	255	255	255	255	255	255	255
R-squared	0.934	0.956	0.939	0.959	0.954	0.968	0.956	0.970

Notes: Robust standard error in brackets. *, ** and *** indicate significance at 10%, 5% and 1% levels, respectively. Each model controls the fixed effect of the industry and year.

**Table 6 ijerph-20-03174-t006:** Regression results after the sample time period is separated into two time periods.

Dependent Variable: *lnCarbon*								
	2000–2008				2009–2014			
Variable	(1)	(2)	(3)	(4)	(5)	(6)	(7)	(8)
*lnDigital*	−0.308 **				−0.518 ***			
	(−2.05)				(−4.04)			
*lnPRODIG*		−0.398 ***		−0.450 ***		−0.526 ***		−0.495 ***
		(−2.70)		(−3.27)		(−3.95)		(−3.89)
*lnDISDIG*			3.718 ***	3.962 ***			2.847 **	2.776 **
			(4.02)	(4.44)			(2.05)	(2.20)
*lnECI*	0.052 **	0.055 ***	0.063 ***	0.071 ***	0.017	0.016	0.014	0.014
	(2.61)	(2.77)	(3.25)	(3.78)	(1.17)	(1.13)	(0.91)	(0.98)
*lnECS*	0.011	0.017	0.038	0.054 **	−0.037 **	−0.032 **	−0.007	−0.008
	(0.41)	(0.61)	(1.40)	(2.02)	(−2.52)	(−2.18)	(−0.38)	(−0.48)
*lnINDS*	−0.001	−0.001	−0.002	−0.002	−0.001	−0.001	0.000	−0.000
	(−0.59)	(−0.60)	(−0.75)	(−0.77)	(−0.58)	(−0.53)	(0.21)	(−0.30)
*lnCLR*	−0.017	−0.020	−0.013	−0.027	−0.006	−0.007	−0.012	−0.007
	(−0.64)	(−0.79)	(−0.53)	(−1.13)	(−0.54)	(−0.62)	(−1.05)	(−0.69)
*lnRDS*	−0.026 *	−0.026 *	−0.011	−0.014	−0.005	−0.005	−0.010	−0.010
	(−1.77)	(−1.80)	(−0.77)	(−1.02)	(−0.68)	(−0.68)	(−1.17)	(−1.36)
*lnFDI*	−0.048 ***	−0.047 ***	−0.061 ***	−0.054 ***	−0.039 *	−0.038 *	−0.059 ***	−0.037 *
	(−3.25)	(−3.23)	(−4.32)	(−3.96)	(−1.85)	(−1.81)	(−2.76)	(−1.82)
*lnERI*	−0.250 ***	−0.247 ***	−0.203 ***	−0.202 ***	0.073	0.080	0.152*	0.133
	(−3.23)	(−3.23)	(−2.72)	(−2.82)	(0.91)	(0.99)	(1.67)	(1.60)
Constant	0.749 ***	0.739 ***	0.614 ***	0.607 ***	0.500 ***	0.473 ***	0.406 ***	0.364 ***
	(7.16)	(7.16)	(5.88)	(6.04)	(5.43)	(5.08)	(3.70)	(3.63)
Industry FE	YES	YES	YES	YES	YES	YES	YES	YES
Year FE	YES	YES	YES	YES	YES	YES	YES	YES
Observations	153	153	153	153	102	102	97	97
R-squared	0.926	0.927	0.932	0.938	0.960	0.960	0.957	0.965

Notes: Robust standard error in brackets. *, ** and *** indicate significance at 10%, 5% and 1% levels, respectively. Each model controls the fixed effect of the industry and year.

**Table 7 ijerph-20-03174-t007:** Regression results of endogenous tests.

Variable	Dependent Variable: *lnCarbon*		
	(1)	(2)	(3)
*L.lnDigital*	−0.542 ***		
	(−3.33)		
*L.lnPRODIG*		−1.050 ***	
		(−4.36)	
*L.lnDISDIG*			7.875 ***
			(7.52)
Constant	0.595 ***	0.557 ***	0.386 ***
	(10.07)	(10.39)	(8.46)
Controls	YES	YES	YES
FE	YES	YES	YES
Kleibergen-Paap rk LM	41.185 ***	41.186 ***	49.616 ***
Kleibergen-Paap rk Wald F	722.331	458.876	414.254
Observations	238	238	238
R-squared	0.979	0.980	0.985

Notes: *** represent significance at the 10% levels, with robust standard errors in parentheses.

**Table 8 ijerph-20-03174-t008:** Mechanism test results for the effects of manufacturing investment in digitization to reduce carbon emission intensity.

Variable	Dependent Variable: *lnCarbon*				
	(1)	(2)	(3)	(4)	(5)
*lnDigital*	−0.378 **	−0.317 *	−0.485 ***	−0.412 ***	−0.386 **
	(−2.34)	(−1.93)	(−3.20)	(−2.70)	(−2.49)
*lnINDS* × *lnDigital*	−0.017 *				
	(−1.90)				
*lnCLR* × *lnDigital*		−0.086 **			
		(−2.48)			
*lnRDS* × *lnDigital*			−0.034		
			(−0.71)		
*lnFDI* × *lnDigital*				−0.038 **	
				(−2.06)	
*lnERI* × *lnDigital*					−0.224 **
					(−2.25)
Constant	0.734 ***	0.733 ***	0.745 ***	0.745 ***	0.740 ***
	(10.53)	(10.58)	(10.64)	(10.73)	(10.68)
Controls	YES	YES	YES	YES	YES
FE	YES	YES	YES	YES	YES
Observations	255	255	255	255	255
R-squared	0.957	0.957	0.956	0.957	0.957

Notes: Robust standard error in brackets. *, ** and *** indicate significance at 10%, 5% and 1% levels, respectively. Each model controls the fixed effect of the industry and year.

**Table 9 ijerph-20-03174-t009:** Heterogeneity analysis results.

Variable	Dependent Variable: *lnCarbon*					
	(1)	(2)	(3)	(4)	(5)	(6)
*lnDigital*	−0.390 **			−0.389 **		
	(−2.44)			(−2.40)		
*lnPRODIG*		−0.818 ***			−0.790 ***	
		(−4.61)			(−4.42)	
*lnDISDIG*			6.510 ***			5.414 ***
			(8.67)			(6.36)
*Clean* × *lnDigital*	−0.160 *					
	(−1.83)					
*Clean* × *lnPRODIG*		−0.144 *				
		(−1.79)				
*Clean* × *lnDISDIG*			1.907 **			
			(2.28)			
*High* × *lnDigital*				−0.151 *		
				(−1.74)		
*High* × *lnPRODIG*					−0.162 **	
					(−2.06)	
*High* × *lnDISDIG*						2.331 ***
						(3.43)
Constant	0.754 ***	0.722 ***	0.487 ***	0.748 ***	0.718 ***	0.524 ***
	(10.83)	(10.80)	(7.65)	(10.75)	(10.80)	(8.60)
Control variables	YES	YES	YES	YES	YES	YES
FE	YES	YES	YES	YES	YES	YES
Observations	255	255	255	255	255	255
R-squared	0.957	0.960	0.969	0.956	0.960	0.970

Notes: Robust standard error in brackets. *, ** and *** indicate significance at 10%, 5% and 1% levels, respectively. Each model controls the fixed effect of the industry and year.

**Table 10 ijerph-20-03174-t010:** Complementary analysis results.

Variable	Dependent Variable: *lnCarbon*		
	(1)	(2)	(3)
*lnDDIG*	−0.770 ***		−0.745 ***
	(−4.19)		(−4.17)
*lnFDIG*		3.934 ***	3.775 ***
		(3.77)	(3.75)
*lnECI*	0.057 ***	0.054 ***	0.055 ***
	(4.45)	(4.17)	(4.36)
*lnECS*	0.031 **	0.013	0.015
	(1.99)	(0.81)	(0.93)
*lnINDS*	−0.003	−0.003 *	−0.004 **
	(−1.61)	(−1.71)	(−2.42)
*lnCLR*	−0.032 *	−0.044 ***	−0.027 *
	(−1.93)	(−2.74)	(−1.71)
*lnRDS*	−0.015	−0.020 *	−0.023 **
	(−1.28)	(−1.69)	(−1.99)
*lnFDI*	−0.064 ***	−0.068 ***	−0.059 ***
	(−5.70)	(−6.01)	(−5.29)
*lnERI*	−0.141 **	−0.052	−0.158 **
	(−2.09)	(−0.82)	(−2.40)
Constant	0.767 ***	0.726 ***	0.768 ***
	(11.01)	(10.47)	(11.36)
Industry FE	YES	YES	YES
Year FE	YES	YES	YES
Observations	255	255	255
R-squared	0.957	0.956	0.959

Notes: Robust standard error in brackets. *, ** and *** indicate significance at 10%, 5% and 1% levels, respectively. Each model controls the fixed effect of the industry and year.

## Data Availability

Not applicable.

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
