# Peer review of "The Effect of Input Digitalization on Carbon Emission Intensity: An Empirical Analysis Based on China’s Manufacturing"

_ijerph, 2023, doi:10.3390/ijerph20043174_

Round 1
Reviewer 1 Report
1. Suggest adding "c" and "independent" in the formula explanation of line 212.
2. Can the order of the second industry classification and the third industry classification in table 1 of line 76 be reversed? What is the meaning of using "other" or "other" in the second industry classification?
3. Is the layout direction of the abscissa years of the four figures in Figure 2 inconsistent? (Figure 2 (a) is different from the other three pictures)
4. Do you want to unify the cell alignment in different tables? (For example, the words "Dependent variable: lnCarbon" in table7 are centered and left-aligned in table8)?
5. Can the gap between Figure 1 (a) input digital distribution and Figure 1 (b) carbon emission intensity distribution ordinate 17 manufacturing industries I1-I17 be larger? Some distributed graphics have exceeded the original interval.
6. In line 344, the chart number in chapter 4.2 is not clear enough.
7.The quality of drawings needs further improvement.
8. In lines 346 to 356, how to improve the description of fluctuation trend of Distributed Input Digitalization, Productive Input Digitalization and Input Digitalization in Figure 2 (a)? For example, add the description of the fluctuation trend of the Distributed Input Digitalization (from the figure, the fluctuation trend of the Distributed Input Digitalization is similar to that of the Productive Input Digitalization, and both are close to that of the Input Digitalization)?
9. 367 in line Add a space between "Figure 3" and "(a)" of "3 (a)"?
10 .Results and Discussions
This section analyzes the results of the data calculation in detail, but the results and the discussion are too integrated. It is suggested to try to split them.
Reviewer 2 Report
I thank the authors for the very problematic research. Their manuscript has the characteristics of an excellent research paper, but I would like to make certain suggestions that the authors can take to improve the manuscript.
I would ask the authors to change the title of the manuscript, so that they do not start the title with a question. I suggest that the title does not contain the form of a question.
The abstract is written very comprehensively, but I ask the authors to check the number of words once more. Certainly, the abstract could be shortened, so that it does not resemble a review of the literature, but rather presents the essence of the problem, the goal, as well as the key results of the research with significant implications.
I will recommend that keywords be monosyllabic words. In the instructions for authors, it is given exactly how to specify keywords.
In the introductory part, the authors provide all the necessary elements for the understanding of the problem, and emphasize the conception of their research. I praise the way of writing the literature review, because all elements are highlighted, it is very meaningful, but I would suggest that the literature review be enriched with more similar research.
The part of the manuscript dedicated to methodology is very rich and comprehensive. All the necessary information about the model is explained clearly for the wider readership. I believe that no changes are needed in the methodology part. All the data and results available to the authors were processed through several models, and each of the models is clearly explained in the methodology part through several subheadings.
The results are presented through tables and figures, very clearly, through 10 tables and 4 figures. The results are explained both through the text and through the necessary tables and figures, which increases the clarity and understanding of the results.
I suggest separating the results from the discussion, if possible.
The conclusions are completely comprehensive, but I suggest that they state more theoretical implications, perhaps even minor ones, because they only stated police implications. To demarcate it. Also, if possible, I suggest that they state the limiting circumstances.
Also, it is necessary to expand the references.
Round 2
Reviewer 1 Report
English language and style are fine/minor spell check required